# Synthetic Analogues of Aminoadamantane as Influenza Viral Inhibitors—In Vitro, In Silico and QSAR Studies

**DOI:** 10.3390/molecules25173989

**Published:** 2020-09-01

**Authors:** Radoslav Chayrov, Nikolaos A. Parisis, Maria V. Chatziathanasiadou, Eleni Vrontaki, Kalliopi Moschovou, Georgia Melagraki, Hristina Sbirkova-Dimitrova, Boris Shivachev, Michaela Schmidtke, Yavor Mitrev, Martin Sticha, Thomas Mavromoustakos, Andreas G. Tzakos, Ivanka Stankova

**Affiliations:** 1Department of Chemistry, South-West University “Neofit Rilski”, 2700 Blagoevgrad, Bulgaria; rchayrov@swu.bg; 2Section of Organic Chemistry and Biochemistry, Department of Chemistry, University of Ioannina, 45110 Ioannina, Greece; nikparis@gmail.com (N.A.P.); m.chatziathanasiadou@gmail.com (M.V.C.); 3Division of Organic Chemistry, Department of Chemistry, National and Kapodistri‘an University of Athens, 15771 Zografou, Greece; evrontaki@pharm.uoa.gr (E.V.); kmoschovou@chem.uoa.gr (K.M.); tmavrom@chem.uoa.gr (T.M.); 4Division of Pharmaceutical Chemistry, Department of Pharmacy, National and Kapodistrian University of Athens, 15771 Zografou, Greece; 5Division of Physical Sciences and Applications, Department of Military Sciences, Hellenic Military Academy, Vari-Koropi Avenue, 16672 Vári, Greece; georgiamelagraki@gmail.com; 6Institute of Mineralogy and Crystallography “Acad. Ivan Kostov”, Bulgarian Academy of Sciences, 1113 Sofia, Bulgaria; sbirkova@mail.bg (H.S.-D.); bls@clmc.bas.bg (B.S.); 7Department of Virology and Antiviral Therapy, Friedrich Schiller University, 207745 Jena, Germany; michaela.schmidtke@med.uni-jena.de; 8Institute of Organic Chemistry with Centre of Phytochemistry, Bulgarian Academy of Science, 1113 Sofia, Bulgaria; yavor@orgchm.bas.bg; 9Department of Analytical Chemistry, Charles University, 11636 Prague 1, Czech Republic; martin.sticha@natur.cuni.cz

**Keywords:** adamantane derivatives, amino acids, 3D-QSAR, molecular docking, X-Ray crystallography, plasma stability, mass spectrometry, LC–MS/MS.

## Abstract

A series of nineteen amino acid analogues of amantadine (Amt) and rimantadine (Rim) were synthesized and their antiviral activity was evaluated against influenza virus A (H3N2). Among these analogues, the conjugation of rimantadine with glycine illustrated high antiviral activity combined with low cytotoxicity. Moreover, this compound presented a profoundly high stability after in vitro incubation in human plasma for 24 h. Its thermal stability was established using differential and gravimetric thermal analysis. The crystal structure of glycyl-rimantadine revealed that it crystallizes in the orthorhombic *Pbca* space group. The structure–activity relationship for this class of compounds was established, with CoMFA (Comparative Molecular Field Analysis) 3D-Quantitative Structure Activity Relationships (3D-QSAR) studies predicting the activities of synthetic molecules. In addition, molecular docking studies were conducted, revealing the structural requirements for the activity of the synthetic molecules.

## 1. Introduction

The synthesis of a series of adamantane derivatives led to the discovery of several drugs that are currently used in clinical practice, such as antivirals, antidiabetics, antimicrobials, anti-inflammatories and CNS-acting compounds [1].

Amantadine (Amt) was discovered in the 60s and it was the first adamantane derivative used in medical practice exhibiting antiviral activity against influenza A strains. Rimantadine (Rim) has been also used since the start of the 1980s to treat and prevent influenza A infection. Both drugs mechanism of action is associated with the proton conduction channel (M2) of influenza A virus. The lack of the M2 ion channel in influenza B virus renders these two drugs ineffective against this virus. There is an increasing number of resistant influenza strains every year as a result of the spontaneous mutations in the viral genome. This fact has led the researchers to the quest for new antiviral drugs that can overcome the resistant virus strains.

One approach that could enhance the antiviral activity of this class of compounds is the incorporation of additional active functional groups in the drug’s scaffold that can disrupt proton transport. Sources of such active functional groups can be amino acids and peptides incorporated into amantadine and rimantadine, using peptide synthesis techniques.

In addition to being the building blocks of proteins and peptides, amino acids serve as precursors of many kinds of small molecules that have important and diverse biological roles. Shibnev et al. [2] studied antiviral activity against influenza A subtypes A (H1N1) and A (H3N2) of various amino acid derivatives of adamantane. It is established that some of the synthesized compounds can inhibit influenza A virus strains resistant to rimantadine. The authors suggest that adamantane carbocycle as a membranotropic carrier is capable of transporting functional groups containing in amino acid to the M2 protein of influenza A/H5N1 virus, thus causing inhibition of influenza A/H5N1 virus replication [3,4].

Hong et al. [5] reported that many oseltamivir carboxylate (A) (neuraminidase inhibitor, NA) analogs have been synthesized (i.e., with lipophilic alkyl and aryl groups other than the 3-pentyl group, different amido groups at the C-4 position or relocating the C–C double bond in the cyclohexene scaffold) bearing anti-influenza activity. However, such structural modifications have not led to enhanced anti-influenza activity with respect to oseltamivir carboxylate (A). In contrast, the replacement of the amino group at the C-5 position in guanidated oseltamivir carboxylate (B) by a more basic guanidino group enhanced the inhibitory activity against NA (Figure 1) [6].

Guanidation has been successfully used in the modification of another class of influenza drugs, the neuraminidase inhibitors—zanamivir (C) [7]. In this frame, derivatization of the synthesized compounds with guanidine could yield even more promising results.

In addition, docking combined with the 3D-Quantitative Structure Activity Relationships (3D-QSAR) results proposes that next generation potential M2 inhibitors should include lipophilic adamantane combined with polar groups, e.g., guanidino group [8].

The aim of this study was the synthesis of amino acid derivatives of amantadine and rimantadine and the investigation of their antiviral activity in vitro against influenza viruses. Some analogues were also guanidated in the amino group. X-ray crystallography was achieved for the analogue with the lowest half maximal Inhibitory Concentration (IC_50_). A plasma stability assay was conducted for this analogue in human plasma, utilizing liquid chromatography coupled with tandem mass spectrometry (LC–MS/MS). Molecular docking combined with the 3D-QSAR study revealed the molecular requirements for drug activities.

## 2. Results and Discussion

### 2.1. Synthesis of the Compounds

The target compounds **2** (**a**–**j**)**, 3a, 3e** and **3g**–**h** were prepared from Boc-amino acids **1** (**a**–**j**) and amantadine or rimantadine using the coupling reagent TBTU (2-(1*H*-Benzotriazole-1-yl)-1,1,3,3-tetramethylaminium tetrafluoroborate) in the presence of DMAP 4-(Dimethylamino)pyridine [9]. Boc-protected groups were removed in CH_2_Cl_2_/TFA at 0 °C and lead to **4** (**a**–**j**)**, 5a, 5e** and **5g**–**h** (Scheme 1).

The derivatives, Ala-Rimantadine (**4a**)**,** β-Ala-Rimantadine (**4i**)**,** Tyr-Rimantadine (**4j**), Ala-Amantadine (**5a**) and Val-Amantadine (**5h**), with the lowest cytotoxicity, were guanidated. Guanidated analogs (**6a**, **6i**–**j**, **7a** and **7h**) were obtained by treating the corresponding aminoadamantane (**4a**, **4i**–**j**, **5a** and **5h**) with 1*H*-pyrazole-1-carboxamidine⋅HCl in 2 mL acetonitrile (Scheme 2) [10].

Boc-(4-F)-Phe-OH was used to modify amantadine since (4-F)-phenylalanine has proven to antiviral activity [11]. Thus, we synthesized and compared the activity between the three amino acids with a similar structure, such as phenylalanine, (4-F)-phenylalanine and tyrosine.

### 2.2. Biological Studies

To improve the oral bioavailability of the antiviral drug acyclovir in 1992, Beauchamp L.M. et al., synthesized eighteen amino acid esters of acyclovir. Among them, L-valyl ester (Valacyclovir) was proved to be a very effective prodrug. Orally introduced in the organism, it possesses 3–4 times higher bioavailability than the acyclovir. Valacyclovir is stable in water–acid solutions, it is swiftly absorbed in vivo at the intestine and subsequently metabolized to acyclovir. According to the literature, the valine derivative is non-toxic, which is also confirmed by our new data for the amantadine amide. The success of valine derivatives against certain viruses, probably due to the amphiphilicity of valine motivated us to couple the structurally similar amino acids leucine and isoleucine with adamantanes [12,13,14].

Their amphiphilic role could play a major role not only in the increased solubility of the new derivatives but also in their facilitated transportation through the cellular and viral membranes [15].

Alanine, one of the most important amino acids, is metabolized in the liver to glucose. This process, known as the glucose-alanine cycle, is reversible and represents one of the main mechanisms for the biochemical synthesis of glucose in the body. Studying the work by Kim et al. [16], we decided to link alanine and amantadine via an amide bond. Alanine exhibits more pronounced hydrophilic properties over the previously described derivatives, which are important for increased bioavailability.

In addition, the structurally similar highly hydrophilic amino acid glycine is very suitable for ameliorating the hydrophobicity of amantadine and can thus, lead to higher bioavailability. This could also be the case for β-alanine. Nonetheless, the use of this amino acid in the synthesis could not be beneficial since it adopts a linear structure and its amino group resembles polyamine. M2 proteins have a specific binding site for polyamines, which is different from the amantadine binding site [17].

Taking into consideration the significance of the nature of the inhibitors in the active sites, we decided to use the guanidino group to modify part of the newly obtained compounds [18]. We chose to modify some of the synthesized derivatives, which showed poor antiviral activity and low cytotoxicity and at the same time to study their biological activity in vitro (Table 1).

As shown in Table 1 the highest antiviral activity of the rimantadine analogs was clearly demonstrated by glycyl-rimantadine (**4b**). Compounds **4d** and **4j** (Leu and Tyr analogues) also displayed antiviral activity but at higher concentrations than **4b** and amantadine. One should note that the IC_50_ of **4b** is about 3.5 times lower than that of amantadine, though with comparable 50% cytotoxic concentration (CC_50_) and thus potentially allowing the use of higher concentration during cell treatment. Probably **4b** small structure allows it to easily fit the M2-ion channel. The branched with aliphatic amino acids rimantadine analogues (Val, Leu, Ile, Ala) do not show antiviral activity. Increasing the volume of the molecule with aromatic amino acids (Phe, 4-F-Phe, Tyr) leads to a decrease or complete disappearance of the antiviral activity. It is important to stress that there is no difference in activity between the R and S isomers **4f** and **4g** and that the different conformational structure of β-Ala (**4i**) greatly decreases the activity.

From the amantadine analogs (**5a**–**h**) only the phenylalanine and 4-F-phenylalanine (**5e**, **5g**) displayed antiviral activity, although at higher concentrations than amantadine. Probably the aromatic ring, as a larger and more rigid structure, does not fit as well in the M2 ion channel.

Modification of amino acid analogs of rimantadine and amantadine with larger and more basic guanidine functional group leads to decreased antiviral activity in contrast to guanidated neuraminidase inhibitors (zanamivir).

### 2.3. 3D-QSAR Studies

The statistical results of the obtained 3D-QSAR comparative molecular field analysis (CoMFA) model for the biologically evaluated molecules are listed in Table 2.

An optimal number of components (ONC) of four corresponds to the lowest standard error of prediction (SEP) of 0.127 and the highest cross-validated Q^2^ of 0.847. A non-cross-validated R^2^ of 0.954, a standard error of estimates (SEE) of 0.070 and the degree of statistical confidence F value of 123.383 show an excellent statistical correlation between the predicted and the actual values of inhibition (Table 2). The CoMFA analysis shows that the relative field contributions were 69.4 and 30.6% for the steric and electrostatic field, respectively. The external validation gave a value of R^2^ test equal to 0.867 upholding an excellent correlation between the predicted and actual values of inhibition.

The results of the CoMFA method were also graphically represented by field contour maps. In these maps the areas around the compound that interacts favorably or unfavorably with the possible receptor are visualized. Therefore, the contour maps are used to identify the structural features relevant to the biological activity of derivatives and their obtained information can be utilized for further design of amino adamantane analogs as blockers of the M2 wild type ion channel.

In Figure 2, the CoMFA fields around compound 29 an active compound from the training set are displayed. The steric interactions are represented by green and yellow contour maps, where bulky groups near the green regions increase the inhibitory activity but they cause the opposite effect near the yellow ones. As it can be observed, the green regions are more than yellow ones for the steric field in the contour map of the CoMFA model (Figure 2a). The large green regions around the aromatic group of the amino adamantane scaffold indicate that bulky groups at o- and m- positions increase the activity, whereas the substitution of bulky groups at p-position of phenyl group decreased the activity. The electrostatic interactions are represented by blue and red contour maps, where electropositive groups were favored near the blue regions, whereas electronegative groups were favored near the red regions, therefore, electropositive groups were favored at m-position and electronegative groups were favored at p-position of phenyl (Figure 2b). Based on the 3D-QSAR CoMFA model, a suggested scaffold of a compound that could be designed and synthesized with potent inhibition against influenza A is depicted in Scheme 3.

The Enalos Model Acceptability Criteria Konstanz Information Miner, (KNIME) node applied to the data shows that the model passed Tropsha’s recommended tests for predictive ability and the results are depicted in Table 3. Using this predictive and validated 3D-QSAR CoMFA model, the predicted inhibitory values of synthesized and evaluated Am analogs from the present study were calculated. The biological activity values (IC_50_) of compounds were compared relatively with the predicted ones (p% inhibition). The active compounds against A/Hong Kong/68 (IC_50_ < 30 μM) were predicted as active (p% > 0.6), and the only inactive one with IC_50_ = 59.71 μM was predicted as inactive (p% = 0.413, Table 4). The correlation plot of the experimental inhibition values versus predicted ones for the 3D-QSAR CoMFA model is shown in Figure 3. It is noticeable that the studied synthesized analogs are *S* enantiomers. R enantiomers were also designed, in order for their inhibition to be predicted.

For continuous QSAR, criteria followed in developing activity/property predictors are: (i) correlation coefficient R between the predicted and actual values; (ii) coefficients of determination (predicted versus observed activities ${\;R}_{0}^{2}$, and observed versus predicted activities ${R^{\prime }}_{0}^{2}$ for regressions through the origin) and (iii) slopes k and  k ′ of regression lines through the origin.

### 2.4. Molecular Docking Study of the Glycyl-Rimantadine

Glycyl-rimantadine (**4b**) showed the lowest IC_50_ (0.11 μM; Figure 4). For this reason, it was sought interest, to dock it inside the M2TM of Influenza A (Figure 5(Β1)). The M2 transmembrane (TM) domain forms a homotetrameric pore and acts as an ion channel by transporting selectively H^+^ from low pH conditions of the endosome into the viral interior [19]. The co-crystallized rimantadine in the M2TM of Influenza A was superimposed with the docked rimantadine using the grid-based ligand docking with energetics (GLIDE) algorithm. The small root mean square deviation (RMSD) (<0.1) of this superposition demonstrates that the use of GLIDE for molecular docking reproduces crystallographic experimental data for structurally similar molecules. In the present molecular docking procedure Rim and glycyl-rimantadine (**4b**) were used as flexible ligands to be docked to the rigid receptor, and XP (extra precision) method was selected in order to calculate the GLIDE score (Figure 5(A1,B1)). The docking calculations showed that glycyl-rimantadine (**4b**) had a lower XP GScore (−7.61 kcal mol^−1^) value than rimantadine’s value (−7.14 kcal mol^−1^), so the binding and the inhibition it causes was expected to be of similar magnitude. The analysis of M2TM–rimantadine interactions (Figure 5(A2)) revealed that the adamantane moiety was oriented to the N-terminus of the virus, while the ammonium group was directed to C-terminus (His37). Furthermore, the adamantane moiety was bounded in a hydrophobic pocket consisting of the residues Val37, Ala30 and Ser31. Specifically, the hydroxyl of Ser31 forms a hydrogen bond to a carbonyl of Val27. It is noteworthy that the bulky adamantane moiety is responsible for shifting the bound water molecules near the top of the pore. As a consequence, the complex is stabilized through hydrophobic interactions (van der Waals) and blocks proton entry into the pore. The ammonium group is adjacent to the four Ala30 carbonyl-associated water molecules. Actually, the ammonium group does not form hydrogen bonds with facing carbonyl groups but it mimics hydrogen that interacts with water networks connected directly to the proton gating residue His37. We examined the intermolecular interactions developed between M2TM and glycyl-rimantadine (**4b**) (Figure 5(Β2)). Two important observations were described. First, both molecules rimantadine and glycyl-rimantadine (**4b**) had their adamantane segment located in the hydrophobic pocket surrounded by the residues Val27, Ala30 and Ser31. Secondly, glycyl-rimantadine (**4b**) adopts an additional hydrogen bond compared to rimantadine with a water molecule, which in a consequence is hydrogen bonding with the key amino acid His37 responsible for the opening of the M2TM channel.

### 2.5. In Vitro Human Plasma Stability of Glycyl-Rimantadine (***4b***)

The evaluation of the stability of glycyl-rimantadine (**4b**) in human plasma was accomplished by setting up the appropriate LC–MS/MS protocol. In order to choose the optimal chromatographic conditions (mobile phase and ratio, column type, flow rate, etc.) for achieving a high resolution and a short runtime, several tests were conducted. A mobile phase composed of formic acid 0.1% in water (A) and acetonitrile (B) along with a reversed-phase C18 column Kinetex (Phenomenex) met those criteria. A gradient elution was chosen for the analysis, at a constant flow rate of 0.25 mL min^−1^ with the following profile: initial phase (B) concentration of 5% until 0.3 min, then increased to 80% within 1.2 min, kept constant for 1 min at 80% and then reduced to 5% at 2.6 min, until the end of the run at 4 min. The temperature in the auto-sampler and the column oven was maintained at 15 °C and 40 °C, respectively. The sample injection volume was set to 5 μL.

Protein precipitation with acetonitrile produced clean chromatograms on blank plasma samples and was therefore chosen as the sample clean-up method. The retention times for the glycyl-rimantadine (**4b**) and Ala-rimantadine (**4a**) (internal standard, IS) were 2.60 min and 2.65 min respectively and the total analysis cycle time was 4 min. Tandem mass spectrometry was used for the detection of the compounds in the positive electrospray ionization (ESI) mode. The tuning parameters (cone gas flow, heated probe temperature, cone temperature, etc.) were optimized by direct infusion of a solution containing glycyl-rimantadine (**4b**) and IS at a concentration of 75 ng mL^−1^ to have a consistent response.

The stability profile of glycyl-rimantadine (**4b**) in human plasma after 24 h incubation at 37 °C is presented in Figure 6. Glycyl-rimantadine (**4b**) was characterized by profoundly high stability in human plasma, since after 6 h the degradation was insignificant (97.5% ± 4.6%). After 12 h approximately 91% ± 3.1% of the conjugate was still present in plasma, while at 24 h of incubation the initial amount was only reduced to 83.7% ± 3.7%.

### 2.6. Elucidation of the Structure of Glycyl-Rimantadine (***4b***)

Single crystal X-ray study showed that the compound **4b** crystallizes in a centrosymmetric manner (space group *Pbca*) as a glycyl-rimantadine trifluoroacetate salt. A summary of the most important crystal data and refinement indicators is provided in Appendix A. The asymmetric unit of the studied compound contains one molecule glycyl-rimantadine, one molecule of trifluoroacetic acid (TFA) and one solvent water molecules (Figure 7). The bond lengths and angles of the glycyl-rimantadine molecule are shown in Appendix A. The X-ray refinement reveals the formation of a zwitterionic TFA^−^–NH_3_^+^-glycyl-rimantadine structure. In support of this fact, the C=O and C–O bond lengths in the TFA^–^ were almost identical (1.231 and 1.242 Å). The crystal structure revealed the presence of several intermolecular hydrogen bonds (Table 5). In the crystal structure, the molecules were oriented head (glycyl) to tail (rimantadine moiety). The bulky rimantadine prevented the formation of hydrogen bonding while the glycyl moiety, TFA and water were all involved in intermolecular interactions. Thus, three-dimensional packing of the structure was governed by the molecular features (hydrophilic and hydrophobic parts) and hydrogen bonding interactions (Figure 8).

### 2.7. Thermal Analysis and Stability of Glycyl-Rimantadine (***4b***)

The thermal behavior of the glycyl-rimantadine (**4b**) compound was investigated from 20 to 250 °C using differential thermal analysis (DTA)—thermogravimetric (TG) analyses (Figure 9). DTA measurement showed two endothermic effects: the first was in the range of 40–70 °C and the second in the range of 150–170 °C. The first *endo* effect was accompanied by 4% weight loss, which corresponded to a loss of one water molecule and is in agreement with the crystal structure. The second DTA effect was not associated with weight loss and corresponded to the melting of the compound. After the melting, a pronounced weight loss was initiated due to the volatility. During the cooling stage, an *endo* effect was observed at around 90 °C.

## 3. Materials and Methods

### 3.1. Chemicals and Reagents

Unless otherwise stated, the starting materials, reagents and solvents were commercially obtained and used as supplied without further purification. Amantadine and rimantadine as hydrochloride salts were from Sigma-Aldrich (St. Louis, Missouri, USA) and the amino acids were purchased from Bachem (Bubendorf, Switzerland). Analytical thin-layer chromatography (TLC) was run on Merck silica gel 60 F-254, with detection by UV light (254 nm). The TLC chromatograms were displayed by ninhydrin solution and Sakaguchi I/II reagents for guanidated derivatives. The reaction was observed by TLC Silicagel 60 F254 (Merck, Darmstadt, Germany) by the following mobile phases: CHCl_3_/CH_3_OH (95:5); CHCl_3_/CH_3_OH/CH_3_COOH (95:5:3) and nBuOH/CH_3_COOH/H_2_O (3:1:1).

^1^H and ^13^C spectra were recorded on Bruker Avance II+ spectrometer (14.09 T magnet, Billerica, Massachusetts, USA) operating at 600.11 MHz 1H frequencies, equipped with a 5 mm BBO probe with z-gradient coil. The temperature was maintained at 293 K, using Bruker B-VT 3000 (v1.9, Bruker, Billerica, Massachusetts, USA) temperature unit with airflow of 535 L h^−1^. All chemical shifts were reported in parts per million (ppm), referenced against tetramethylsylane (TMS, 0.00 ppm) or using the residual solvent signal (7.27 ppm for CDCl_3_ of 2.5 ppm for DMSO).

Electrospray mass spectrometry (ESI-MS) experiments were acquired on Bruker Compact QTOF-MS (Bruker Daltonics, Billerica, Massachusetts, USA) and controlled by the Compass Control software (v1.9, Bruker, Billerica, Massachusetts, USA). The data analysis was performed and the monoisotopic mass values were calculated using Data analysis software (v4.4, Bruker, Billerica, Massachusetts, USA). The analyses were conducted in the positive ion mode at a scan range from *m*/*z* 50 to 1000 and nitrogen was used as nebulizer gas at a pressure of 4 psi and flow of 3 L min^−1^ for the dry gas. The capillary voltage and temperature were set at 4500 V and 220 °C, respectively. An external calibration for mass accuracy was carried out by using sodium formate as calibration solution. The precursor ion of each compound was selected, and ESI–MS/MS analysis was performed by collision-induced dissociation (CID); nitrogen was the collision gas, and the collision energy varied from 5 to 40 eV. MSn experiments were conducted on an ion trap instrument Esquire 3000 (Billerica, Massachusetts, USA) and controlled by the Esquire Control software (v5.3.11, Bruker, Billerica, Massachusetts, USA). ESI-MS data were collected in positive-ion mode at a scan range from *m*/*z* 50 to 500. In all ESI-MS measurements, the nebulizer gas pressure was 124.1 kPa at a flow rate of 5 L min^−1^; the desolvation temperature was 300 °C and capillary voltage was adjusted to 4000 V. The sample solutions were delivered to nebulizer by a syringe pump Cole Parmer (Chicago, Illinois, USA) at a flow rate 3 µL min^−1^.

Formic acid (98%, LC–MS grade) and dimethyl sulfoxide (DMSO, ≥99.7%) were obtained from Honeywell Fluka (Charlotte, North Carolina, USA) and Thermo Scientific (Waltham, Massachusetts, USA), respectively. LC–MS grade acetonitrile and water were purchased from Thermo Fisher Scientific. Membrane filters (0.2 μm pore size and 47 mm diameter, Waltham, Massachusetts, USA and Minisart RC 4 syringe filters (0.2 μm pore size) were purchased from Sartorius (Göttingen, Germany Drug-free, pooled human plasma from healthy donors was a kind offer from the Blood Donation Center of the University Hospital of Ioannina (Ioannina, Greece).

General Procedure for Synthesis of the **2**(**a**–**j**), **3a**, **3e** and **3g**–**h**.

The Boc-AA (**1**(**a**–**j**) (3 mmol), DIPEA (3.1 mmol) were added to a solution of TBTU (3 mmol) in CH_2_Cl_2_ (15 mL). After stirred the mixture was treated with amantadine or rimantadine (3 mmol) with DMAP (3 mmol). This mixture was stirred at RT for 3 h, and then evaporated to dryness. After evaporation the residue was purified by flash chromatography (ethyl acetate:n-hexan, 1:1) as the eluent.

#### 3.1.1. General Procedure for Boc-Group Deprotection of the **4**(**a**–**j**), **5a**, **5e** and **5g**–**h**

One equiv. of **2**(**a**–**j**), **3a**, **3e** and **3g**–**h** was dissolved in 10-fold excess of trifuoroacetic acid (TFA) at 0 °C. The reaction mixture was stirred until Boc group running out (the chromatographic control was carried out in systems chloroform-methanol 95:5 ratio). The solvent was evaporated and the residue was dissolved in 10 mL methanol. Until the pH reached approximately 8, 25% ammonia solution was added. The solvent was evaporated under vacuum. Obtained crystals was dissolved in ethyl acetate and washed with water 3 × 25 mL. The organic layer dried on anhydrous Na_2_SO_4_ and solvent was removed. The yield on each compound was various, but it was within 65–82%.

#### 3.1.2. Preparative Method of the **6a**, **6i**, **6j**, **7a** and **7h**

In 2 mL acetonitrile, 0.67 mmol of **4a**, **4i**, **4j**, **5a** and **5h** was dissolved. Of 1*H*-pyrazole-1-carboxamidine hydrochloride 1.0 mmol and 2.0 mmol TEA were added to the solution. The reaction continued for 48 h at room temperature. After that the solvent was evaporated under vacuum. The residue was dissolved in 25 mL chloroform and washed few times with 5% sodium hydrogensulfate (pH 3). The organic layer was dried with sodium sulfate anhydrous. The solvent was evaporated under vacuum and residue was crystalized by methanol/diethyl ether. After 12 h at 4 °C the mixture was filtered and dried at least 24 h in a desiccator, filled with calcium dichloride.

### 3.2. Cytotoxicity

The synthesized compounds were tested for antiviral activity against the influenza virus A/H3N2, strain Hong Kong 68.

In order to identify new potential antiviral drugs, small amounts of extracts or compounds have to be examined for cytotoxicity and antiviral activity in primary screening using a rapid, easy, inexpensive and highly standardized test system. The cytotoxic and the antiviral effects were quantified using a crystal violet uptake assay allowing automated handling of large numbers of candidate agents. To ensure comparable results with plaque reduction assays, the 50% plaque inhibitory concentrations of amantadine and oseltamivir were used to standardize the anti-influenza virus A tests. Madin-Darby Kanine Kidney Cells (MDCK) cells were seeded at 2 × 10^4^ cells per well in 96-well flat-bottomed microtiter plates (Falcon 3075). The cytotoxicity of the test compounds was determined on 2-day-old confluent MDCK cell monolayers grown in the internal 60 wells (5% CO_2_, 37 °C). After removal of the growth medium, two-fold dilutions of the compounds in 100 µL test medium were added and incubated at 37 °C in a 5% CO_2_ atmosphere for 72 h. Cells in six wells without compound treatment served as basic controls. The Dynex Immuno Assay System (DIAS, Guernsey, UK) developed for automated ELISA techniques was applied to wash gently, stain, measure and analyze the viability of the cell monolayers in cytotoxic, as well as antiviral assays. The staining procedure undertaken with crystal violet was based on the principles described by Nain et al. for the determination of the cytotoxic activity of TNF-α [20]. At first, the supernatant was aspirated and the cell monolayers were washed three times with 300 mL physiological sodium chloride solution to remove dead cells. Secondly, the cells were fixed and stained in one step with 50 µL of 0.03% crystal violet (*w*/*v*) in 20% methanol for 10 min. After six further automated washings with 300 µL of water, the stained monolayers were treated for 20 min with 100 µL lysis buffer (0.8979 g of sodium citrate and 1.25 mL of 1 N HCl in 98.05 mL of 47.5% ethanol) to elute the crystal violet. Then the optical density of individual wells was quantified spectrophotometrically at 540/630 nm and analyzed with the DIAS. Cell viability was evaluated as the percentage of the mean value of optical density, resulting from the six cell controls, which was set 100%. The 50% cytotoxic concentrations (CC_50_) were calculated from the mean dose–response curves of three assays each with two parallels.

### 3.3. Antiviral Activity

To exclude non-specific antiviral activities of test compounds, their cytotoxic effects have to be determined not only on confluent cell monolayers but also on proliferating cell cultures in the primary antiviral screening. The cytotoxicity and the growth inhibition of compounds were quantified using an automated crystal violet staining procedure and the electronic cell analyzer system CASY 1, respectively, on day 3 after compound application. The 50% cytotoxic and growth inhibition concentrations were calculated from the mean dose–response curve (3.9 through 1000 µg mL^−1^; dilution factor 2) of three separate experiments each with two parallels in the indicated cell lines.

### 3.4. 3D-QSAR Model Development

For the development of the 3D-QSAR model, 38 molecules were used with experimental inhibitory data obtained from the literature [21,22]. The selected structures of amino adamantane analogs were built using ChemBioDraw Ultra 14.0 [23] and their SMILE strings were imported in the project table of Maestro10.2, the graphical user interface (GUI) by Schrödinger [24]. 2D chemical structures were converted to 3D using LigPrep3.4, [25] in which all the hydrogen atoms were added, ionization states were generated at user-defined pH 7, and the molecules were geometrically refined. All the structures were subsequently submitted in full structure optimization, using the minimization procedure of MacroModel 10.8 [26]. For the minimization, water was selected as the solvent and a standard molecular mechanics energy function (OPLS_20057 force field) with the Polak–Ribiere conjugated gradient method (PRCG, 1000 iterations with gradient 0.01 kcal mol^−1^ Å^−1^) was applied [27]. The dataset was divided into two sets, training and test set, followed by the module previously described [28]. Therefore, the training set consisted of 28 molecules and the test one consisted of the remaining 10 molecules. The experimental data of 38 adamantane analogs expressed as the percent inhibition (%), ranging from 0.7 to 91, were converted into (×10^−2^) values and are defined in Appendix A.

### 3.5. CoMFA Analysis

The CoMFA calculations were performed using SYBYL 8.0 software provided by TRIPOS [29]. Atomic charges of molecules were calculated using the Gasteiger–Hückel method, which is a combination of two other charge computational methods: the Gasteiger–Marsili [30] method that calculates the σ component of the atomic charge, and the Hückel [27] method that calculates the π component of the atomic charge. The total charge is the sum of the charges calculated by the two methods. A rigid-body atom-by-atom superimposition of one molecule onto another was performed using the amino adamantane core (11 atoms) as the common substructure, and the alignment of compounds was applied based on amantadine (Figure 10).

The CoMFA fields are generated by creating a grid around the molecule and calculating the steric and electrostatic potentials at each point on the grid using a + 1 charged probe atom of a hybridized sp^3^ carbon [31,32]. The steric and electrostatic field energies were calculated using the Lennard-Jones and the Coulomb potentials, respectively, with a 1/r^2^ distance-dependent dielectric constant in all intersections of regularly spaced (0.2 nm) grid. The truncation for both the steric and electrostatic energies was set to 30 kcal mol^−1^. This indicates that any steric or electrostatic field value that exceeds this value will be replaced with 30 kcal mol^−1^, thus makes a plateau of the fields close to the center of any atom.

### 3.6. Partial Least Square (PLS) Analysis

The partial least square (PLS) analysis [33,34] applied in SYBYL 8.0 was used to derive the 3D-QSAR model and linearly correlate the CoMFA fields with experimental values. The CoMFA fields were used as independent variables and the inhibitory values as dependent variables. PLS analysis was performed in two steps. In the first step, the cross-validation analysis performed using leave-one-out (LOO) method was applied to the training set of the dataset in order to determine (Q^2^), SEP and ONC. In the LOO method, one compound is omitted from the training set and the model is derived involving the rest of the molecules. Then using this model, the activity of the molecule that was removed can be predicted [32]. The model with the lowest SEP and as possible highest Q^2^ was selected for the next step of the analysis. The value of Q^2^ is calculated by the following equation:(1)$$Q^{2}=1-\frac{\sum^{\;}{\left(Y_{\mathrm{obs}}-Y_{\mathrm{pred}}\right)}^{2}}{\sum^{\;}{\left(Y_{\mathrm{obs}}-{\bar{Y}}_{\mathrm{train}}\right)}^{2}}$$
where $Y_{\mathrm{obs}}$ and $Y_{\mathrm{pred}}$ indicate observed and predicted activity values, respectively, and ${\bar{Y}}_{\mathrm{train}}$ indicates the mean of observed activity values of the training set. A model is considered acceptable when the value of Q^2^ exceeds 0.5 [35,36].

In the second step, using the ONC obtained from cross-validation analysis, a non-cross-validated method was applied and the non-cross-validated R^2^ (conventional) was calculated according to the following equation:(2)$$R^{2}=1-\frac{\sum^{\;}{\left(Y_{\mathrm{obs}}-Y_{\mathrm{calc}}\right)}^{2}}{\sum^{\;}{\left(Y_{\mathrm{obs}}-\bar{Y}\right)}^{2}}$$

In addition, the statistical significance of the model is described by the standard error of estimate (SEE) and the probability value (F-value). To speed up the analysis and reduce noise, a minimum column filter value σ (2.00 kcal mol^−1^) was used. The application of non-cross-validated method led to the build of the CoMFA model.

### 3.7. 3D-QSAR CoMFA Model Validation

The predictive capability of the 3D-QSAR CoMFA model was evaluated using the test set [37]. The predictive correlation (R^2^_test_) is calculated using the following equation:(3)$$R_{\mathrm{test}}^{2}=1-\frac{\sum^{\;}{\left(Y_{\mathrm{pred}\left(\mathrm{test}\right)}-Y_{\mathrm{test}}\right)}^{2}}{\sum^{\;}{\left(Y_{\mathrm{test}}-{\bar{Y}}_{\mathrm{train}}\right)}^{2}}$$
where $Y_{\mathrm{pred}\left(\mathrm{test}\right)}$ and $Y_{\mathrm{test}}$ indicate predicted and observed activity values of the test set, respectively. For a predictive QSAR model, the value of the R^2^_test_ should be more than 0.5.

To assess the predictive power of the produced 3D-QSAR model, the Enalos Model Acceptability Criteria KNIME node was additionally used [38] including all proposed validation tests [39].

### 3.8. Molecular Docking of the Glycyl-Rimantadine (***4b***) in the Binding Site of Rimantadine Bound to the Influenza A M2 Transmembrane Domain

Molecular docking is one of the most preferable methods in drug discovery design because of its ability to predict the conformation of small molecule ligands within the protein binding site with accuracy. In addition, the most likely binding conformation and the corresponding intermolecular interactions can be determined.

The ligands (rimantadine and glycyl-rimantadine) were built using ChemBioDraw Ultra, [22] and their SMILES used in order to create 3D-structure using Maestro 10.2 Graphical User Interface (GUI) [23]. Then, the two ligands were minimized by means of Maestro MacroModel 10.8, [25] where all hydrogens were added and molecules were subjected to complete structure minimization. For the minimization, water was chosen as a solvent, OPLS_2005 [26] force field and the algorithm Polak-Ribiere (PRCG, convergence value 0.01 kcal mol^−1^ Å^−1^) were used. Subsequently, the molecules were checked to have the right pH (7.4) using LigPrep 2.5. [40].

The 3D-structure of Influenza A M2TM (open conformation)–rimantadine (PDB code: 6BOC) served as a model structure for the docking calculations. The “Protein Prep. Wizard” application from Schrödinger’s Maestro software was used, which is a collection of tools that prepares the 3D-structure of proteins [40,41,42,43].

Using the “Preprocess and analyze structure” tool, the sequence of the bonds was checked, all hydrogen atoms were added, all disulfide bonds were checked, and all water molecules greater than 5 Å from the active site of the protein were deleted. Epik 2.3 was used in order to predict the ionization of amino acids [40,41,42,43].

Finally, using the “Impref” application and the OPLS_2005 force field optimized, the positions of the enzyme hydrogen atoms by keeping all other atoms immobilized [23].

For the molecular docking was used Schrödinger’s GLIDE (grid-based ligand docking with energetics) algorithm [44]. The binding region was defined by a 12 Å × 12 Å × 12 Å grid whose center coincides with that of the ligand and selected the binding of ligands similar in size to the crystallized ligand. The other parameters used were the program default. The precision level XP (XP extra precision) and XP GScore were used for docking and grading, respectively. The poses representing the lowest value of the scoring function (XP GScore) were further analyzed to identify which of the ligand conformations most likely describe the correct binding mode.

### 3.9. LC–MS/MS Plasma Stability Assay

#### 3.9.1. LC–MS/MS Conditions

The separation of glycyl-rimantadine (**4b**) and Ala-rimantadine (**4a**) (internal standard, IS) was performed on an ultra-high-performance liquid chromatography (UHPLC) system (Bruker Daltonics, Germany), utilizing an RP Kinetex C18 column 100 mm × 2.1 mm, 2.6 μm, with a proguard column 2.1 mm (Phenomenex, USA). The mobile phase was composed of formic acid 0.1% in water (A) and acetonitrile (B). Gradient elution was chosen for the analysis, at a constant flow rate of 0.25 mL min^−1^ and the following gradient profile: initial phase (B) concentration of 5% until 0.3 min, then increased to 80% within 1.2 min, kept constant for 1 min at 80% and then reduced to 5% at 2.6 min, until the end of the run at 4 min. The temperature in the autosampler and the column oven was maintained at 15 °C and 40 °C, respectively. The sample injection volume was set to 5 μL.

The EVOQ Elite ER (Bruker Daltonics, Germany) triple quadrupole mass spectrometer was operated in positive ionization electrospray mode (ESI) in multiple reaction monitoring (MRM) for the detection of glycyl-rimantadine (**4b**) and IS. The optimal MRM transitions for monitoring the analytes were determined and set as follows: *m*/*z* 237.1 → 163.2, 237.1 → 75.5 and 237.1 → 81.4 for glycyl-rimantadine and 251.1 → 163.2, 251.1 → 89.4 and 251.1 → 180.2 for the IS. The optimum ESI parameters were determined and set as follows: spray voltage at 4500 V; heated probe gas flow at 50 units; heated probe temperature at 150 °C; cone gas flow at 20 units; cone temperature at 300 °C and nebulizer gas flow at 50 units. The control of the LC and the MS, as well as the data acquisition, was performed using the MSWS software, version 8.2.1 (Bruker Datonics, Germany).

#### 3.9.2. Stock, Working, Calibration Solutions and Extraction Procedure

Stock solutions (1 mg mL^−1^ and 100 μg mL^−1^) and working solutions for the glycyl-rimantadine **(4b)** and the IS were prepared in DMSO. Five working solutions with concentrations equal to 4, 10, 20, 30 and 40 μg mL^−1^ were used in order to prepare the final calibration standards (10, 25, 50, 75 and 100 ng mL^−1^) for the analyte. The final concentration of the IS was 50 ng mL^−1^. Protein precipitation was applied to extract the analyte from plasma. Specifically, for each calibrator 5 μL of the respective working solution were spiked in 90 μL of plasma. Then, 400 μL of ice-cold acetonitrile were used for plasma precipitation and 5 μL of the IS working solution (20 μg mL^−1^) were added. Vigorous vortexing and centrifugation for 10 min at 10,000× *g* were employed to separate and collect the liquid supernatant containing the analyte and the IS. The supernatant was filtered with 0.2 μm syringe filters and 100 μL were transferred in LC–MS vial containing 300 μL water (LC–MS grade). Finally, 5 μL were injected into the LC–MS system. The calibration standards for the conjugate were freshly prepared on the experimental days by spiking the appropriate amounts of working solutions into the blank human plasma. All stock and working solutions were stored at −20 °C prior to analysis.

#### 3.9.3. In Vitro Plasma Stability

The in vitro plasma stability of glycyl-rimantadine (**4b**) was conducted by LC–MS/MS. In 90 μL of blank human plasma, 5 μL of the respective working solution (30 μg mL^−1^) were added and incubated for 0, 2, 4, 6, 12 and 24 h in a thermoshaker at 37 °C. At the end of the incubation time the samples were processed as described above. All samples were tested in triplicates. The glycyl-rimantadine (**4b**) concentration was calculated via the standard calibration curve that was designed by plotting the area of the conjugate versus the IS area against the nominal concentration of the calibration standards. The plot of the remaining percentage of the parent compound against time was designed for the glycyl-rimantadine analogue through GraphPad Prism 8.0.1 (Bruker, Billerica, Massachusetts, USA).

### 3.10. Thermal Analyses

DTA and TG thermal analysis were carried out simultaneously in a thermal analyzer Stanton Redcroft STA780 at the following conditions: heating rate of 5 °C min^–1^, dry argon as carrier gas (30 mL min^–1^) and sample weight of 7 mg in corundum crucible.

### 3.11. Crystal Structure Analysis

#### 3.11.1. Sample Crystallization

In order to obtain crystals with better quality 50 mg of the dry powder (microcrystals) of glycyl-rimantadine trifluoroacetate (C_16_H_27_N_2_O_4_F_3_) were dissolved in ethanol, methanol and acetone (3 mL) at room temperature. Large crystals (0.3 × 0.2 × 0.15 mm^3^) suitable for single crystal X-ray studies grew by slow evaporation within 2–3 days from all solutions, however the diffraction quality was significantly better for the crystals obtained from methanol solution. Crystals were collected directly from the mother liquors, e.g., from the three solvents used.

#### 3.11.2. Data Collection and Crystal Structure Refinement

Selected crystal was mounted on a glass capillary and all data were collected at room temperature (290 K) on an Oxford Diffraction SupernovaDual four cycle diffractometer, equipped with Atlas CCD detector using Mo-Kα radiation (λ = 0.71013 Å) from the micro-focus source. The determination of unit cell parameters, data integration, scaling and absorption corrections were carried out using the CrysalisPro. [45]. The phases were obtained by direct methods with ShelxS-2018/3. The refinement of the structure involved several cycles of refinement using full-matrix least-squares on *F^2^* with the ShelxL-2018 package [46,47]. Initially non-hydrogen atoms (C, O, N and F) were located and positioned from difference Fourier map. After that, all hydrogen atoms were located and positioned in the resulting difference Fourier map. The non-hydrogen atoms were refined anisotropically and hydrogen atoms were refined using the riding model. The drawings of the asymmetric unit and three-dimensional representations of glycyl-rimantadine (**4b**) were carried out using *ORTEP* [48] and Mercury programs [49]. The crystallographic data (coordinates and structure factors) have been verified using check CIF/PLATON [50].

## 4. Conclusions

The antiviral activity in vitro against the influenza virus A/H3N2 strain Hong Kong/68 of new analogs of amantadine and rimantadine conjugated with amino acids was investigated.

The results revealed that the highest antiviral activity combined with low cytotoxicity was demonstrated by the rimantadine derivative with the structurally simplest amino acid-glycine. IC_50_ of the conjugate was about 3.5 lower than amantadine. Moreover, glycyl-rimantadine (**4b**) shows a high stability profile after incubation in human plasma for 24 h. Docking studies disclosed that rimantadine and glycyl-rimantadine (**4b**) had their adamantane segment located in the same hydrophobic pocket. Furthermore, glycyl-rimantadine adopted an additional hydrogen bond compared to rimantadine with a water molecule, which in a consequence was hydrogen bonding with the key amino acid His37.

In addition, the various aliphatic and aromatic amino acids as substitutes in the rimantadine molecule had no significant effect on antiviral activity. The corresponding amantadine analogs were considerably less effective and only phenylalanyl-amantadine (**5e**) and (4-F)-phenylalanyl-amantadine (**5g**) exhibited activity although lower than that of the amantadine in the same concentration. The results indicate that electron-acceptor substituent (4-F)-phenylalanine did not change the activity. Probably the aromatic ring, as a larger structure, did not fit in the M2 ion channel.

QSAR studies on adamantane analogues revealed that the CoMFA model could predict the activities of the synthetic molecules.

Amantadine and rimantadine analogues conjugated with guanidine show low toxicity but they also exhibited lower activity. Unlike guanidated oseltamivir analogues [6] the guanidation of the rimantadine and amantadine analogues did not lead to increasing of antiviral activity.

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
