# Peer review of "Synthetic Analogues of Aminoadamantane as Influenza Viral Inhibitors—In Vitro, In Silico and QSAR Studies"

_molecules, 2020, doi:10.3390/molecules25173989_

Round 1

Reviewer 1 Report

please, check attached file

Author Response

цж

Thank you very much for the reviewers remarks, comments and questions.

We have attempted to improve the manuscript and below are listed the corrections and the answers to the reviewers.

Reviewer1  comments:

Comment 1. The Table on the pages 6 and 7 has no number, title or description of data included. Please add missing description and reference in the text of article.

Answer: The table has been removed from the page 5 as it is listed as Table S2 in the 2.6. Elucidation of the Structure of Glycyl-rimantadine, and is present in the SI.

Comment 2. Abstract: “CoMFA (Comparative Molecular Field Analysis) 37 3D QSAR studies predicting the activities of synthetic molecules and proposing the synthesis of 38 analogues with R stereochemistry as potential molecules to possess higher activity.”

This assumption is mainly based on computational, predicted data based on 3D QSAR studies, it wasn’t proved with experimental data. In my opinion, without experimental confirmation/comparison of R and S derivatives, this information is slightly useless and not complete.

Answer: We would like to thank the reviewer for the constructive comment. We have deleted the sentence “and proposing the synthesis of analogues with R stereochemistry as potential molecules to possess higher activity.” 

Comment 3. The Scheme 1, although a general one, is to simplified and hardly readable in my opinion. I think, that more detailed structures are required. Please find below my suggestion (editable in ChemDraw) and consider revision.

Answer: Scheme 1 was changed.

Comment 4. Please number all structures (substrates and products) throughout this publication.

Answer: All structures were numbered.

Comment 5. It’s not necessary to present description of NMR spectra in both the main article and supplementary file. Please, choose one. If you decide to keep it in the main article and remove it from supplementary file, place it in the ‘Materials and Methods’ Section.

Answer: The NMR description has been removed from the manuscript and is now present only in the SI.

Comment 6. Figure 1. Compounds (A), (B), (C) (D) are not mentioned/referred in the text, please make the references.

Answer: Figure 1 and the compounds heve been refeenced in the manuscript on page 2:  

“However, such structural modifications have not led to enhanced anti-influenza activity with respect to oseltamivir carboxylate. In contrast, the replacement of the amino group at the C-5 position in guanidated oseltamivir carboxylate by a more basic guĐ°nidino group enhanced the inhibitory activity against NA (Figure 1) [7]”.

Comment 7. In the Introduction part, there is no any information about possible modifications/derivatives of amantadine or rimantadine. I only found a brief historical description of parent compounds. Such information could be more useful then description of oseltamivir analogues. I understand, that this isn’t the review about antiviral adamantine derivatives but some information should be included. At least, you could present modifications of parent compounds, which potentiate the antiviral effect. A kind of ‘starting point’ is required.

Answer: In the Introduction part some additional data and references have been added.

Comment 8. In the experimental part please add in what form the chemical compound was obtained (oil, solids, crystals, color) and give the melting point for solids.

Answer: It's done.

Comment 9. Highlight the used synthetic methods in the experimental part, add the general method/conditions for the synthesis of guanidine derivatives.

Answer: It's done.

Comment 10. Line 366, Figure 4. Figure is to small and hardly readable. Please, use bigger and more sharpen Figure.

Answer We have updated and enlarged Figure 4 in order to be more readable.

Figure 4. The two-dimensional (2D) and three-dimensional (3D) structures of rimantadine and Glycyl-rimantadine

Comment 11. 659 In order to obtain crystals with better quality 50 mg of the dry powder (microcrystals) of Glycyl 660 rimantadine (C16H27N2O4F3) were dissolved in ethanol, methanol and acetone (3 mL) at room…

Based on the empirical formula should be glycyl-rimantadine trifluoroacetate (salt).

Line 420, page 14. In the table the glycyl-rimantadine has an empirical formula C16H27N2O4F3 which suggest a trifluoroacetate salt. Please, clarify.

Answer: The reviewer is correct and the crystals obtained are for a glycyl-rimantadine trifluoroacetate. We have modified the sentence:

“Single crystal X-ray study showed that the compound X crystallizes in a centrosymmetric manner (space group Pbca) as a Glycyl-rimantadine trifluoroacetate salt.”

Comment 12. page 21, 660 rimantadine (C16H27N2O4F3) were dissolved in ethanol, methanol and acetone (3 mL) at room…

For crystallization process you used a mixture of three solvents or performed three separated crystallizations from each solvent? Please indicate the best (successful) solvent.

Answer: In the 3.10.1 (Sample crystallization) we  improved the  description for  crystal growth.

“Large crystals (~ 0.3 x 0.2 x 0.15 mm3) suitable for single crystal X-ray studies grew by slow evaporation within 2-3 days from all solutions, however the diffraction quality was significantly better for the crystals obtained from methanol solution.”

Comment 13. I suggest to move Table 5 to supplementary file.

Answer: Table 5 has been moved to the SI (as Table S1). The numbering of the SI in the manuscript has been adjusted accordingly.

Comment 14. The Scheme 2, although a general one, is to simplified and hardly readable in my opinion. I think, that more detailed structures are required. Please find below my suggestion (editable in ChemDraw) and consider revision.

Answer: Scheme 2 was changed.

Comment  15. References’ section: Only the first word in the title of the article should be capitalized, please correct references: 1, 8, 17, 28, 29, 31, 32, 34,

Comment  16. References’ section: the journal abbreviation should include period, for example: Curr. Med. Chem. not Curr Med Chem

Comment  17. References’ section: please, use the correct journals abbreviations:
ref 5 – Tetrahedron Lett.
ref 8 - Antivir. Chem. Chemother.
ref. 24 = J. Enzyme Inhib. Med. Chem.
ref. 27 - J. Chem. Eng. Data
ref. 30, 33 – J. Chem. Inf. Comput. Sci.

Answer: We have checked carefully the References and corrected all remarks listed in points 15-17.

Reviewer 2 Report

The authors of “Synthetic Analogues of Aminoadamantane as Influenza Viral Inhibitors Ë— In Vitro, In Silico and QSAR studies” made a interesting work, which is suitable for its publication in Molecules. Nevertheless, they need to make some corrections and some specifications.

  • The adamantane derivatives works by two action mechanisms, which depends on its concentration. The basic nature of the adamantanes changes the optimal pH for the viral replication, in one of the mechanisms, and blocks the matrix (M2) protein, for the other mechanism. In this work, it is clear that the authors study in a depth the blocking ability of M2 protein by the aminoadamantane, but what can you say about the basic nature of these compounds?
  • They need to justify the use of logarithmic normalization of the experimental values used in the QSAR model.
  • Some of the parameters for the statistical validation of the QSAR model are lacking, like standard deviation.
  • Where is the QSAR mathematical model? they need to show it to help the understanding of the structural parameters important to the activity.
  • They need to explain which type of molecular docking they carried out, rigid or flexible? Which aa were considered if the docking was flexible.
  • They need to improve their figures of the docking results. Also, they can incorporate 2D representation of the intermolecular interactions between aminoadamantane and M2 protein.

Author Response

Thank you very much for the reviewers remarks, comments and questions.

We have attempted to improve the manuscript and below are listed the corrections and the answers to the reviewers.

Reviewer 2

Comment 1. The adamantane derivatives works by two action mechanisms, which depends on its concentration. The basic nature of the adamantanes changes the optimal pH for the viral replication, in one of the mechanisms, and blocks the matrix (M2) protein, for the other mechanism. In this work, it is clear that the authors study in a depth the blocking ability of M2 protein by the aminoadamantane, but what can you say about the basic nature of these compounds?

Answer: Additional data and references on the nature of these compounds are added to the Introduction part.

Comment 2. They need to justify the use of logarithmic normalization of the experimental values used in the QSAR model.

Answer We used logarithmic normalization of the experimental values as the software we used (SYBYL 8.0 software provided by TRIPOS) for the 3D-QSAR model development requires values from 0.0-1.0, the most inactive is expressed by the value 0.0 and the most active by the value 1.0. Thus, for the specific software we used is absolutely required the logarithmic scale.

Comment 3. Some of the parameters for the statistical validation of the QSAR model are lacking, like standard deviation.

Answer We want to thank the reviewer for this comment. Indeed, there are more statistical parameters for the validation of a QSAR model (e.g. SD, P-value, RMSE, R- Pearson), nevertheless the used in the present study statistical technique exports only the specific parameters that are listed in Table 4, which constitute the specifications for characterizing it as a predictive or not predictive model. Thus, again the specific software limits and guides the statistical parameters to export and use for the prediction of the obtained model as stable or not.

Comment 4. Where is the QSAR mathematical model? they need to show it to help the understanding of the structural parameters important to the activity. Specifically, the mode

Answer The model is a 3-dimensional one.  Therefore no linear function can be extracted with the structural parameters as the independent variables and the inhibition as dependent one that occurs in the 2-dimensional models. The results of the CoMFA method are represented by using 3D contour maps. That is the reason it is characterized as 3D model and not 2D model that is based on the mathematical equations giving statistical weight on the dependence parameters. This is the great advantage of the 3D methods. They provide a visualization of the results that guide immediately medicinal chemist to make structural modifications and synthesize structures with optimized biological activity.

The results of the CoMFA method were also graphically represented by field contour maps. In these maps the areas around the compound that interacts favorably or unfavorably with the possible receptor are visualized. Therefore, the contour maps are used to identify the structural features relevant to the biological activity of derivatives and their obtained information can be utilized for further design of amino adamantane analogs as blockers of M2 wild type ion channel.”

Comment 5. They need to explain which type of molecular docking they carried out, rigid or flexible? Which aa were considered if the docking was flexible.

Answer We thank the reviewer for the constructive comment. We have explained the molecular docking method we have carried out in the revised version.

In the present molecular docking procedure Rim and glycyl-rimantadine were used as flexible ligands to be docked to the rigid receptor, and XP (Extra Precision) method was selected in order to calculate the Glide Score. “

Comment 6 .They need to improve their figures of the docking results. Also, they can incorporate 2D representation of the intermolecular interactions between aminoadamantane and M2 protein.

Answer The figures are improved and 2D representation of the intermolecular interactions between aminoadamantane and M2 protein are included.

Figure 5. A1. The 3D docked pose of M2TM – rimantadine  in  the crystallized M2TM (PDB code: 6BOC) with the presence of water molecules (red color). Hydrogen bonds are depicted as yellow dashes; A2. The 2D docked pose of M2TM – rimantadine interactions within 5Å.   B1. The 3D docked pose of M2TM – glycyl-rimantadine  D. The M2TM – rimantadine interactions, B2. The 2D docked pose of M2TM – glycyl-rimantadine interactions within 5Å.  

Thank you very much for the reviewers remarks, comments and questions.

We have attempted to improve the manuscript and below are listed the corrections and the answers to the reviewers.

Reviewer 2

Comment 1. The adamantane derivatives works by two action mechanisms, which depends on its concentration. The basic nature of the adamantanes changes the optimal pH for the viral replication, in one of the mechanisms, and blocks the matrix (M2) protein, for the other mechanism. In this work, it is clear that the authors study in a depth the blocking ability of M2 protein by the aminoadamantane, but what can you say about the basic nature of these compounds?

Answer: Additional data and references on the nature of these compounds are added to the Introduction part.

Comment 2. They need to justify the use of logarithmic normalization of the experimental values used in the QSAR model.

Answer We used logarithmic normalization of the experimental values as the software we used (SYBYL 8.0 software provided by TRIPOS) for the 3D-QSAR model development requires values from 0.0-1.0, the most inactive is expressed by the value 0.0 and the most active by the value 1.0. Thus, for the specific software we used is absolutely required the logarithmic scale.

Comment 3. Some of the parameters for the statistical validation of the QSAR model are lacking, like standard deviation.

Answer We want to thank the reviewer for this comment. Indeed, there are more statistical parameters for the validation of a QSAR model (e.g. SD, P-value, RMSE, R- Pearson), nevertheless the used in the present study statistical technique exports only the specific parameters that are listed in Table 4, which constitute the specifications for characterizing it as a predictive or not predictive model. Thus, again the specific software limits and guides the statistical parameters to export and use for the prediction of the obtained model as stable or not.

Comment 4. Where is the QSAR mathematical model? they need to show it to help the understanding of the structural parameters important to the activity. Specifically, the mode

Answer The model is a 3-dimensional one.  Therefore no linear function can be extracted with the structural parameters as the independent variables and the inhibition as dependent one that occurs in the 2-dimensional models. The results of the CoMFA method are represented by using 3D contour maps. That is the reason it is characterized as 3D model and not 2D model that is based on the mathematical equations giving statistical weight on the dependence parameters. This is the great advantage of the 3D methods. They provide a visualization of the results that guide immediately medicinal chemist to make structural modifications and synthesize structures with optimized biological activity.

The results of the CoMFA method were also graphically represented by field contour maps. In these maps the areas around the compound that interacts favorably or unfavorably with the possible receptor are visualized. Therefore, the contour maps are used to identify the structural features relevant to the biological activity of derivatives and their obtained information can be utilized for further design of amino adamantane analogs as blockers of M2 wild type ion channel.”

Comment 5. They need to explain which type of molecular docking they carried out, rigid or flexible? Which aa were considered if the docking was flexible.

Answer We thank the reviewer for the constructive comment. We have explained the molecular docking method we have carried out in the revised version.

In the present molecular docking procedure Rim and glycyl-rimantadine were used as flexible ligands to be docked to the rigid receptor, and XP (Extra Precision) method was selected in order to calculate the Glide Score. “

Comment 6 .They need to improve their figures of the docking results. Also, they can incorporate 2D representation of the intermolecular interactions between aminoadamantane and M2 protein.

Answer The figures are improved and 2D representation of the intermolecular interactions between aminoadamantane and M2 protein are included.

Figure 5. A1. The 3D docked pose of M2TM – rimantadine  in  the crystallized M2TM (PDB code: 6BOC) with the presence of water molecules (red color). Hydrogen bonds are depicted as yellow dashes; A2. The 2D docked pose of M2TM – rimantadine interactions within 5Å.   B1. The 3D docked pose of M2TM – glycyl-rimantadine  D. The M2TM – rimantadine interactions, B2. The 2D docked pose of M2TM – glycyl-rimantadine interactions within 5Å.   

Round 2

Reviewer 1 Report

please, see attached file

Author Response

The authors would like to thank reviewer for his valuable time taken to evaluate the manuscript, and for pointing out improvement possibilities.

The corrections are given in red color.